# Improving Soluble Expression of SARS-CoV-2 Spike Priming Protease TMPRSS2 with an Artificial Fusing Protein

**DOI:** 10.3390/ijms241310475

**Published:** 2023-06-22

**Authors:** Xiao Ye, Xue Ling, Min Wu, Guijie Bai, Meng Yuan, Lang Rao

**Affiliations:** 1National Technology Innovation Center of Synthetic Biology, Key Laboratory of Engineering Biology for Low–Carbon Manufacturing, Tianjin Institute of Industrial Biotechnology, Chinese Academy of Sciences, Tianjin 300308, China; yexiao1996@163.com (X.Y.); lingxue@tib.cas.cn (X.L.); wum@tib.cas.cn (M.W.); baigj@tib.cas.cn (G.B.); yuanm@tib.cas.cn (M.Y.); 2Key Laboratory for Molecular Enzymology and Engineering, The Ministry of Education, School of Life Science, Jilin University, Changchun 130012, China

**Keywords:** TMPRSS2, SARS-CoV-2, artificial protein, protein expression, protease

## Abstract

SARS-CoV-2 relies on the recognition of the spike protein by the host cell receptor ACE2 for cellular entry. In this process, transmembrane serine protease 2 (TMPRSS2) plays a pivotal role, as it acts as the principal priming agent catalyzing spike protein cleavage to initiate the fusion of the cell membrane with the virus. Thus, TMPRSS2 is an ideal pharmacological target for COVID-19 therapy development, and the effective production of high–quality TMPRSS2 protein is essential for basic and pharmacological research. Unfortunately, as a mammalian–originated protein, TMPRSS2 could not be solubly expressed in the prokaryotic system. In this study, we applied different protein engineering methods and found that an artificial protein XXA derived from an antifreeze protein can effectively promote the proper folding of TMPRSS2, leading to a significant improvement in the yield of its soluble form. Our study also showed that the fused XXA protein did not influence the enzymatic catalytic activity; instead, it greatly enhanced TMPRSS2′s thermostability. Therefore, our strategy for increasing TMPRSS2 expression would be beneficial for the large–scale production of this stable enzyme, which would accelerate aniti–SARS-CoV-2 therapeutics development.

## 1. Introduction

Severe acute respiratory syndrome coronavirus 2 (SARS-CoV-2), which caused the ongoing COVID-19 pandemic, remains a public health threat. SARS-CoV-2 belongs to the same coronavirus family as SARS-CoV, and both viruses infect the target cells in similar manners, as the spike (S) protein of the virus binds to cell surface receptor angiotensin–converting enzyme 2 (ACE2) and stimulates the fusion of the viral lipid envelope with cellular membranes [1]. The S protein needs to be primed, a proteolytic cleavage process exposing the S protein’s membrane fusion domain, before initiating membrane fusion. Furin and TMPRSS2 are enzymes that perform proteolytic cleavage and prime the S protein. In this process, Furin cleaves the S protein to release the S1 and S2 subunits, and TMPRSS2 further cleaves the S2 subunit (S2′), which is located immediately upstream of the hydrophobic fusion peptide, triggering the exposure of the membrane fusion domain hydrophobic peptide (FP) [2,3,4]. Because of its essential role in S2’ cleavage, an indispensable requirement for exposing the FP, TMPRSS2 is considered a critical regulator for spike–mediated viral evasion [5].

TMPRSS2, a membrane protein consisting of 492 amino acids, belongs to the type II transmembrane serine protease family [6]. It is expressed in a wide range of human tissues, including the prostate, liver, and lung, with its original physiological functions related to digestion, inflammation, and tumor invasion [7]. Notably, TMPRSS2 deletion in mice does not result in a major pathological condition, suggesting that the gene may be non–essential for the organism’s development [8]. In contrast, TMPRSS2–deficient mice showed significantly reduced pathological severity when infected by MERS–CoV [9] and SARS-CoV-2 [10]. Therefore, TMPRSS2 is a good target for developing pan–coronavirus antiviral therapy because a potent TMPRSS2 inhibitor might efficiently inhibit coronavirus infection without having any significant adverse effects on the human body. The effective production of TMPRSS2 protein is an essential prerequisite for studying basic and pharmacological aspects concerning TMPRSS2. Until now, TMPRSS2 has been expressed heterogeneously within different systems, including yeast [11], mammalian cells [12], and insect cells [13]. However, there are few reports on the expression of TMPRSS2 in prokaryotic expression systems such as *Escherichia coli* due to the production of insoluble protein [14].

XXA is a recently developed protein solubilizing fusion tag [15]. XXA has a reversed amino acid sequence of the antifreeze protein from *Chlorella sorokiniana*, (Appendix A), AXX, making it a retro–protein of AXX and also classifying it as an artificial protein since its amino acid sequence does not exist naturally. With high solubility and hydrophilicity, XXA exerts good properties in facilitating the refolding of inclusion bodies, and it outperforms other tags such as SUMO, MBP, and GST when tested with proteins such as protease bdNEDP1 and nanobody NbALFA [15].

In this study, we used different strategies to optimize the expression of soluble TMPRSS2 in *E. coli* and found that the artificial fusion protein XXA had an overwhelming effect on facilitating the correct folding and expression of TMPRSS2.

## 2. Results

### 2.1. XXA Fusion Protein Increased the Expression of Soluble TMPRSS2 in E. coli

The full–length TMPRSS2 protein is composed of four major domains: the low–density lipoprotein receptor class A domain, the scavenger receptor cysteine–rich domain, the serine protease domain, and the transmembrane helical domain (Appendix A). As multiple pieces of evidence suggest that the transmembrane domain does not contribute to the enzymatic activity of the protein [14,16,17], we chose a truncated TMPRSS2 (106–492) for our study. The *E. coli* codon–usage–optimized DNA of TMPRSS2 was synthesized de novo into the expression vector pET28a. The recombinant plasmid pET–His−TMPRSS2 was then transformed into *E. coli* BL21 (DE3) cells and induced with various IPTG concentrations. The cell lysates were separated into soluble and insoluble fractions and analyzed using SDS–PAGE. Under varied inducing conditions, we were able to identify a distinct protein band in the insoluble part of the cell lysate but not in the soluble cell lysate, which corresponded to the theoretical molecular weight (42.8 kDa) of TMPRSS2 (Figure 1). These results, therefore, confirmed the previous findings that when expressed in *E. coli,* the recombinant TMPRSS2 protein remained as an insoluble inclusion body [18].

To enhance the solubility of the recombinant TMPRSS2 in our *E. coli* system, we chose three common solubility enhancer tags: small ubiquitin–related modifier (SUMO) [19], maltose binding protein (MBP) [20,21], and glutathione–S–transferase (GST) [22]. Plasmids coding for the fusion proteins SUMO–TMPRSS2, MBP–TMPRSS2, and GST–TMPRSS2 were constructed (Appendix A) and transfected into *E. coli* BL21 (DE3), and the induction of protein expression was carried out at 16 °C to minimize inclusion body formation. However, the fusion of three common solubility tags failed to improve the solubility of recombinant TMPRSS2, as SDS–PAGE analysis of the induced cell lysates showed the presence of the protein bands with the corresponding molecular weight only in the insoluble cell pellet (Figure 2a). These results showed that the conventional protein solubilization methods had limited effect on improving the expression of soluble TMPRSS2.

Artificial protein XXA, developed by Ning Shi’s lab, is a newly invented solubilizing fusion tag that exerts excellent performance in improving the solubility of inclusion bodies [15]. To assess the effect of XXA on the solubility of TMPRSS2, we generated a plasmid construct, pET–XXA–TMPRSS2, in which the XXA tag was fused to the N–terminal of TMPRSS2 (Appendix A). The expression of this construct was induced in *E. coli*, and the resulting cell lysates were analyzed. Our studies revealed a prominent protein band in the soluble cell lysate that corresponded to the molecular weight of XXA–TMPRSS2 (67 kDa) (Figure 2b). It should be noted that under the same inducing condition (16 °C, IPTG 0.6 mM), the expression of soluble XXA–TMPRSS2 was substantially greater than that of the other fusion proteins (Figure 2a). Furthermore, when the protein was induced at a higher temperature (25 °C) and with a higher IPTG dosage (1 mM), the majority of XXA–TMPRSS2 was still expressed in a stable soluble form (Appendix A). This suggested that the XXA fusion tag could stabilize the protein in multiple conditions.

To compare the efficacy of all these fusion tags in enhancing the expression of soluble TMPRSS2, we measured the activity of crude cell lysates from all transfected cells because the proteolytic activity of the sample would reflect the active form of the enzyme. A typical protease fluorogenic substrate, Boc–Gln–Ala–Arg–AMC, was chosen for the enzymatic assay. The results indicated that the MBP, GST, and SUMO tags led to a 2–3 fold increase in soluble TMPRSS2 yield compared to the His tag (Figure 2c). However, the XXA tag showed a significant advantage in promoting the active TMPRSS2 expression, resulting in a 16–fold increase in proteolytic activity (Figure 2c). This result was consistent with the prominent XXA–TMPRSS2 protein band found in the soluble cell lysate (Figure 2b).

### 2.2. Purification and Identification of XXA–TMPRSS2

To characterize the enzymatic properties of the recombinant TMPRSS2, we purified the recombinant protein XXA–TMPRSS2 via Ni–chelating affinity chromatography. The purified protein was then analyzed via SDS–PAGE, and a single protein band, consistent with the theoretical molecular weight of XXA–TMPRSS2 (67 kDa), was visualized (Figure 3a). We then performed immunoblotting for the purified recombinant protein using an anti–His tag antibody and observed a specific band on the blotting membrane, which showed the correct expression of the recombinant protein (Figure 3b).

To further confirm that the protein expressed was indeed XXA–TMPRSS2, we excised the corresponding gel band after SDS–PAGE and digested the protein into peptides, which were then analyzed via chromatography–mass spectrometry (MALDI–TOF). Mass spectrometry analysis revealed multiple peptide sequences matching those of TMPRSS2 (Figure 4). These data conclusively demonstrated that TMRPSS2 could be soluble when expressed in the prokaryotic expression system *E. coli*. These results confirmed that XXA is an ideal fusion tag that significantly improves the solubility of TMPRSS2.

### 2.3. The Fusing Protein XXA Does Not Influence Catalytic Efficiency of TMPRSS2

To investigate the effect of the fused XXA tag on TMPRSS2, we compared the enzymatic activity of TMPRSS2 with and without the XXA tag. We generated TMPRSS2 without tag by first incubating XXA–TMPRSS2 with HRV 3C protease, thereby allowing proteolytic cleavage between the XXA tag and TMPRSS2. We then applied the reaction complex to a Ni column in which the cleaved XXA–His protein bound to the Ni resin, allowing TMPRSS2 without the fusion tag to flow through the column and be collected for further study. The purified TMPRSS2 without any tag was designated as ΔXXA–TMPRSS2 (Figure 5a). Both XXA–TMPRSS2 and ΔXXA–TMPRSS2 were used for different biochemical assays. We first investigated the effect of pH on TMPRSS2 by incubating the enzymes in reaction systems with different pH. The results revealed that the overall pH profiles of the two enzymes were largely similar; both enzymes were active in the pH range of 7.5 to 11.0, with an optimal pH of 8.5. However, XXA–TMPRSS2 performed better in an alkaline environment from pH 9 to 10 (Figure 5b).

The kinetic parameters of XXA–TMPRSS2 and ΔXXA–TMPRSS2 were determined via non–linear regression based on the Michaelis–Menten equation. The K_m_, *k*_cat_, and V_max_ values for XXA–TMPRSS2 were 1.6 μM, 0.82 s^−1^, and 8.23 nmol/min, respectively, while those for ΔXXA–TMPRSS2 were 1.7 μM, 0.93 s^−1^, and 9.3 nmol/min, as shown in Figure 6. The K_m_ values for XXA–TMPRSS2 and ΔXXA–TMPRSS2 differed slightly but were consistent with previous studies [18]. The calculated *k*_cat_/K_m_ ratio for XXA–TMPRSS2 was slightly lower than that of ΔXXA–TMPRSS2 (0.51 μM^−1^s^−1^ vs. 0.54 μM^−1^s^−1^ ). These results demonstrate that the XXA tag has a negligible impact on the catalytic activity of TMPRSS2.

### 2.4. XXA Tag Improves the Thermostability of TMPRSS2

To examine how the XXA tag affected the stability of TMPRSS2, XXA–TMPRSS2, and ΔXXA–TMPRSS2 were incubated at 45 and 55 °C for different durations, and the thermal stability of the enzymes was calculated by measuring the residual enzymatic activity of the heat–treated enzymes (Figure 7). We found that XXA–TMPRSS2 maintained more than 80% activity after 4 h at 45 and 55 °C, whereas ΔXXA–TMPRSS2 lost more than 80% of its original activity. It thus suggested that the XXA tag improved the thermostability of TMPRSS2.

To further investigate the impact of XXA on thermostability, we conducted a comparative analysis of the thermal denaturation processes of XXA–TMPRSS2 and ΔXXA–TMPRSS2 using circular dichroism (CD) spectroscopy. To ensure the proper folding of the proteins in their native form at 20 °C, we first scanned the entire UV range of the CD spectrum for both proteins (Appendix A).

To monitor the denaturation of the proteins during heating from 20 to 100 °C, we selected a CD absorption wavelength of 220 nm, which is characteristic of the α–helical protein secondary structure. As the temperature increased, both proteins underwent denaturation, resulting in a loss of their secondary structure, as evidenced by changes in ellipticity measured using CD spectroscopy. Specifically, XXA–TMPRSS2 began to denature at approximately 70 °C, as indicated by a dramatic decrease in CD intensity (Figure 8a). On the other hand, ΔXXA–TMPRSS2 started denaturing at around 40 °C, as observed by a change in CD intensity (Figure 8b). Both proteins were almost completely inactivated at 80 °C. Based on the measured sigmoidal unfolding curve, the melting temperatures (Tm) of the following two enzymes were calculated: XXA–TMPRSS2 had a Tm of 73.8 °C, and ΔXXA–TMPRSS2 had a Tm of 67.7 °C (Figure 8). These results further demonstrate that the thermal stability of XXA–TMPRSS2 is superior to that of ΔXXA–TMPRSS2. Therefore, we conclude that the XXA tag does not affect the enzymatic properties of TMPRSS2 and actually improves its thermal stability.

## 3. Discussion

It is well established that TMPRSS2 plays a role not only in infectious diseases but also in cancer progression. Recent evidence shows a critical role of TMPRSS2 in processing the viral spike surface protein, thereby facilitating the transmission of multiple coronaviruses [14,23,24]. This role of TMPRSS2 in viral transmission makes it a potential target for developing drugs to treat multiple coronaviruses. Thus, obtaining the purified TMPRSS2 protein is crucial for the discovery of enzymatic inhibitors. However, few reports have confirmed the successful expression of TMPRSS2 using a prokaryotic protein expression system such as *E.coli*, even though expressing proteins in *E.coli* has substantial cost and time benefits over eukaryotic expression systems. The biggest challenge for expression in *E. coli* is the inability to express TMPRSS2 in a soluble form. There have been several attempts to express TMPRSS2 in *E.coli*, but in all those cases, the protein was expressed as an insoluble inclusion body [14,18]. To obtain active TMPRSS2 protein from inclusion bodies, the aggregates need to be denatured and then refolded via rapid dilution in a refolding buffer. This protein refolding technique is difficult and time–consuming and necessitates the use of complex dialyzer equipment [18].

To avoid inclusion body formation and facilitate soluble protein expression, Mahoney et al. fused TMPRSS2 with a pelB leader sequence to deliver it to the periplasm, an environment with protein chaperones that promotes folding, resulting in soluble TMPRSS2 expressed in *E. coli* [24]. However, the periplasmic expression has limitations, such as low productivity due to cellular toxicity caused by the protein produced [25,26]. Additionally, the process of extracting protein from periplasm is more laborious than that of harvesting it from the cytoplasm [26].

In this study, we aimed to produce high levels of soluble and active TMPRSS2 in the cytoplasm using the *E. coli* expression system. We first tried three typical soluble fusion partners, GST, MBP, and SUMO, but found out that those tags do not affect TMPRSS2′s solubility, as we could barely find any TMPRSS2 protein in the soluble fraction of the cell lysates (Figure 2A). Then, we decided to use XXA, an artificial protein that has a reversed amino acid sequence of the antifreeze protein AXX from *C. sorokiniana*. Xi Xie et al. first discovered the solubilizing properties of AXX and then found that its retro–protein XXA has even better solubilizing effects. Thus, Xi Xie et al. developed XXA as a protein–solubilizing expression fusion tag [15]. With the fusing tag XXA, the soluble TMPRSS2 yields increased five times compared with the GST tag. Our work demonstrated the unbeatable advantage of XXA as a fusion tag for improving TMPRSS2 protein folding compared to other well−established fusion tags, as most of the XXA−fused TMPRSS2 was expressed in soluble form (Figure 2B).

A desirable solubility fusion tag should enhance the solubility of the target protein while preserving its physical and chemical properties. Our results showed that the XXA tag did not adversely affect the enzymatic properties, such as optimal pH preference and kinetic efficiency. Thus, it is not necessary to remove the fusion tag of the recombinant protein in subsequent enzymatic assays, which could save the cumbersome process of tag removal. Since the discovery of XXA–mediated protein folding, our work is the second successful attempt at utilizing this fusion tag for protein expression and highlights the potential of XXA as a robust solubility–enhancing tag. Our research also demonstrated that, in addition to facilitating solubility, the XXA tag acts as a protein stabilizer, improving the thermostability of TMPRSS2 (Figure 7 and Figure 8). The improved thermostability is a bonus that XXA brought to TMPRSS2, which would benefit the enzyme’s storage and application.

In this study, we discovered a novel way to overcome the expression problem of TMPRSS2 in *E. coli* by fusing it with an artificial protein XXA. The fusion tag not only facilitated the correct folding but also improved the thermostability of the enzyme without compromising its biological activity. The soluble expression of TMPRSS2 in *E. coli* would facilitate downstream research on TMPRSS2–specific inhibitors discovery and ultimately promote COVID-19 therapy development.

## 4. Materials and Methods

### 4.1. E. coli Strains and Reagents

*E. coli* TransT1 and *E. coli* BL21 (DE3) were obtained from TransGen (Beijing, China). DNA polymerase, DNA ladder, and protein markers were from Solarbio (Beijing, China). Exonuclease, HRV 3C Protease, and DNA ligase were from Takara (Dalian, China). Boc–Gln–Ala–Arg–AMC was from Bachem (Bubendorf, Switzerland). 6–isopropyl–β–d–thiogalactopyranoside (IPTG) was from Sigma–Aldrich (St. Louis, MO, USA). His antibody was from Beyotime (Shanghai, China). All other chemicals were of the highest reagent grade commercially available.

### 4.2. Amino Acid Sequence

XXA: KLRDAADQAAKSADGALDEGKAQARGLGEKADGKLESYKEKATDAVEAHRRAEDAAEAGKLGERAAGQADRGAGEAGGAADRVAREASGGLESATSKAGEAAESARQKAEYAEQAAAKDGLTAAKQEAYGLNQRQDQTVDRATEQVDAAAGTVTEKVKQAADSVAHKATEIASKAKDALSEDQM

### 4.3. Gene Cloning and Protein Expression

DNA encoding TMPRSS2 (106−492) were codon optimized for *E. coli* expression and synthesized by GENEWIZ Bio Inc. (Suzhou, China). The Gene coding recombinant protein His−TMPRSS2, MBP−TMPRSS2, SUMO−TMPRSS2, GST−TMPRSS2, and XXA–TMPRSS2 were synthesized and cloned into protein expression plasmids with primers as listed below with standard gene cloning procedures (Table 1). 

*E. coli* TransT1 was utilized for gene cloning manipulation and plasmid propagation. All recombinant plasmids were sequenced to confirm the correct insertion of the target DNA before subsequent usage. To express the recombinant proteins, the plasmids were transfected into *E. coli* BL21 (DE3) and grown in Luria–Bertani (LB) medium at 37 °C with shaking at 220 rpm. When OD600 value reached 0.6–0.8, protein expression was induced by the addition of 0.6–1 mM IPTG followed by another 16 h of growth at either 16 °C or 25 °C. Finally, the induced cells were harvested via centrifugation at 5000× *g* for 10 min at 4 °C.

### 4.4. Protein Purification and XXA Tag Removal

The collected cells were resuspended in Tris–HCl buffer (20 mM, pH 8.0) and disrupted via ultrasonication on ice for 10 min. After ultrasonication, the supernatant was collected via centrifugation at 15,000× *g* for 15 min. Then, the supernatant was loaded onto a Ni–chelating affinity chromatography column (GE Healthcare, Chicago, IL, USA). The chromatography column was washed with binding buffer (50 mM Tris–HCl, 300 mM NaCl, pH 8.0) and washing buffer (50 mM Tris–HCl, 300 mM NaCl, pH 8.0, 40 mM imidazole) to remove nonspecific protein. The targeted protein was collected by washing the column using elution butter (50 mM Tris–HCl, 300 mM NaCl, pH 8.0, 250 mM imidazole). The obtained protein solution was desalted, concentrated, and stored at −80 °C for further analysis. To remove the XXA tag from the recombinant protein XXA–TMPRSS2, the protein was first diluted to a concentration of 1 mg/mL in 1 × HRV 3C buffer. HRV 3C protease (1–5 U) was then added to the mixture, and the solution was incubated at 4 °C for 16 h. This allowed the protease to cleave off the XXA tag from the protein. The resulting reaction mixture was then loaded onto a Ni–chelating affinity chromatography column, where the XXA tag, now bound to Ni with the His tag adjacent to it. The Δ XXA–TMPRSS2 protein was then collected in the flow–through fraction.

### 4.5. SDS–PAGE and Western Blotting

Protein concentration was measured spectrophotometrically using Bradford color–reaction assay kit (Bio–Rad, Hercules, CA, USA) with bovine serum albumin as standard SDS–PAGE was performed using 5% stacking gel and 10% resolving gel, and proteins were stained with Coomassie brilliant blue R–250. SDS–PAGE separated proteins were transferred onto a polyvinylidene fluoride membrane using a Mini Trans–Blot Transfer Tank. The membrane was then briefly washed with PBS buffer, following a 1 h incubation with 5% skim milk to block the membrane. An appropriately diluted primary antibody anti–His was incubated with the membrane for 2 h, followed by washing with TBST three times. The membrane was then incubated with secondary antibodies, followed by three more washes with TBST. After the final washing, the membrane was stained with the SuperECLTM Plus Western Blotting Detection Kit (US EVERBRIGHT INC, Sayreville, NJ, USA). The chemiluminescence signal was detected using a Tanon 5200 chemiluminescent imaging system.

### 4.6. Mass Spectrometry

The band of XXA–TMPRSS2 was excised from the SDS–PAGE gel and placed in an Eppendorf tube. The protein in gel pieces was reduced with 10 mM DTT at 97 °C for 10 min and alkylated with 40 mM 2–chloroacetamide at room temperature for 30 min. They were then dried with acetonitrile and digested with 30 ng/μL trypsin in 50 mM ammonium bicarbonate at 37 °C overnight. The digested peptides were dried under vacuum and reconstituted in a 0.1% (*v*/*v*) formic acid (FA) solution. Subsequently, the peptides were then separated within 30 min at a flow rate of 350 nL/min on a homemade column (75 μm × 25 cm, 1.9 μm C18−AQ particles, Dr. Maisch, Frankfurt, Germany). Mobile phase A consisted of 0.1% (*v*/*v*) FA in H2O, and mobile phase B consisted of 0.1% (*v*/*v*) FA in 100% acetonitrile. Peptides were separated with a gradient of 5–95% mobile phase B. All peptide samples were analyzed using a hybrid TIMS quadrupole time–of–flight mass spectrometer (Bruker timsTOF Pro2) equipped with a CaptiveSpray nanoelectrospray ion source. The mass spectrometer was operated in DDA–PASEF mode. The MS data were processed with MaxQuant v1.6.2.0, and the MS spectra were compared to the theoretical amino acid sequence of XXA–TMPRSS2.

### 4.7. Enzymatic Activity Assay

Boc–Gln–Ala–Arg–AMC (Boc is t–Butyloxy carbonyl, and AMC is 7–amino–4–methylcoumarin) was chosen as substrate. The proteolytic activity was measured by monitoring the release of fluorescent compound AMC from the substrate. The reaction mixture comprised 2 μL of Boc–Gln–Ala–Arg–AMC (10 μΜ), 83 μL of reaction buffer (50 mM Tris pH 8.0), and 5 μL of appropriately diluted enzyme solution. The mixture was then incubated for 1 h at room temperature, and the release of the fluorescent compound AMC was monitored using a Microplate Reader (BioTek, Winooski, VT, USA) with 360 nm excitation and 460 nm emission. The fluorescence intensity was recorded at 3 min intervals over a period of 60 min.

To study the impact of pH on XXA–TMPRSS2 and ΔXXA–TMPRSS2, the activity of the enzymes was measured in reaction buffers with pH ranging from 3.0 to 11.0 in different buffers: 20 mM glycine–HCl (pH 3.0–5.0), Na_2_HPO_4_–NaH_2_PO_4_ (pH 5.0–7.0), Tris–HCl (pH 7.0–9.0), and Gly–NaOH (pH 9.0–11.0). The stability of the enzymes was determined by measuring their residual activity after incubating them in PBS (20 mM, pH 7.5) at 45 °C and a concentration of 0.2 mg/mL.

### 4.8. Enzyme Kinetic Studies

For the enzyme kinetic studies, purified XXA–TMPRSS2 and ΔXXA–TMPRSS2 mixed with various concentrations of Boc–Gln–Ala–Arg–AMC as the substrate (ranging from 0 to 12.28 μM) were measured at room temperature pH8.0. Kinetic parameters K_m_, *k*_cat_, and V_max_ were determined via nonlinear regression fitting of the data to the Michaelis–Menten equation via the GraphPad Prism 9 software.

### 4.9. Circular Dichroism (CD) Analysis

The CD spectra were acquired using a temperature–ramped Chirascan CD spectropolarimeter (Applied Photophysics Limited, Leatherhead, UK). The thermal denaturation was analyzed by measuring far–UV CD spectra with a heating rate of 1 °C per minute, covering temperatures ranging from 20 to 100 °C. The purified XXA–TMPRSS2 and ΔXXA–TMPRSS2 were diluted in a PBS buffer (20 mM, pH 7.5) to a final concentration of 1 mg/mL and placed in a quartz cuvette with a 0.5 mm path length. Tm values were determined by monitoring temperature–induced changes in the CD signal at 220 nm and calculated using Global 3 software (Applied Photophysics Limited, Leatherhead, UK).

## Figures and Tables

**Figure 1 ijms-24-10475-f001:**
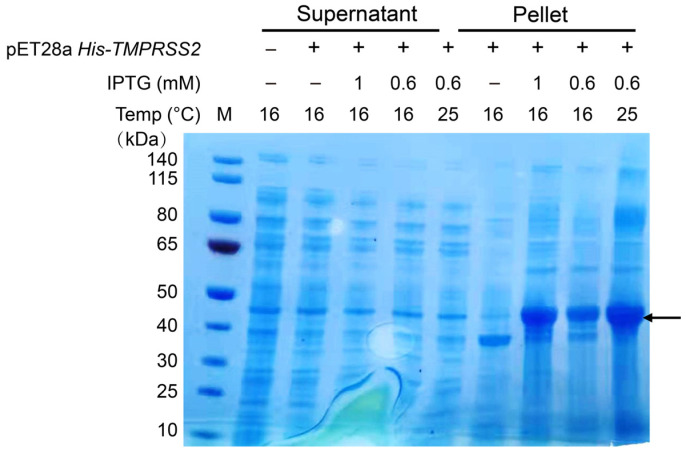
TMPRSS2 expressed as inclusion body. *E. coli* BL21 (DE3) was transfected with pET28a–His−TMPRSS2 and induced for 16 h with 0.6 or 1 mM IPTG at 16 or 25 °C. The cellular extract was separated into soluble supernatant and insoluble pellet fractions and analyzed via SDS–PAGE. The black arrow indicates the recombinant protein His−TMPRSS2.

**Figure 2 ijms-24-10475-f002:**
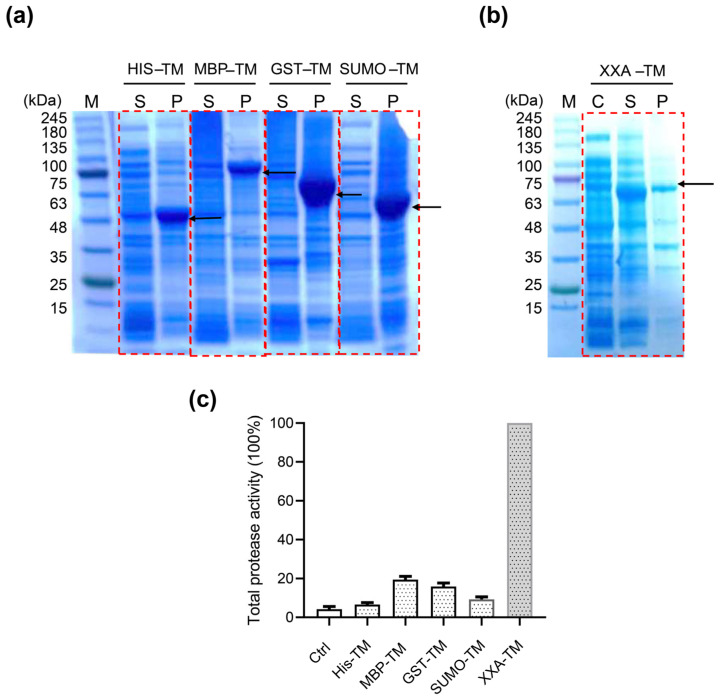
XXA tag promotes soluble expression of TMPRSS2. *E. coli* BL21(DE3) transfected with pET–His–TM, pET–MBP–TM, pET–SUMO–TM, pGEX–GST–TM, (**a**) and pET–XXA–TMPRSS2; (**b**) the above TM refers to TMPRSS2 and then induced with IPTG (1 mM) for 16 h at 16 °C. The soluble cell lysate (S) and insoluble pellets (P) were separated via centrifugation at 8000 rpm. pET–XXA–TMPRSS2 transfected cells without IPTG induction were chosen as negative control (C). The red box highlights the cell lysate from cells transfected with each plasmid, and the black arrows indicate the recombinant TMPRSS2 protein. (**c**) The proteolytic activity was examined with the crude soluble cell lysates using fluorescent trypsin substrate [Boc–Gln–Ala–Arg–AMC]. The relative activity of the enzyme was calculated considering the maximum activity as 100%. The error bar represents the standard deviation of two independent assays.

**Figure 3 ijms-24-10475-f003:**
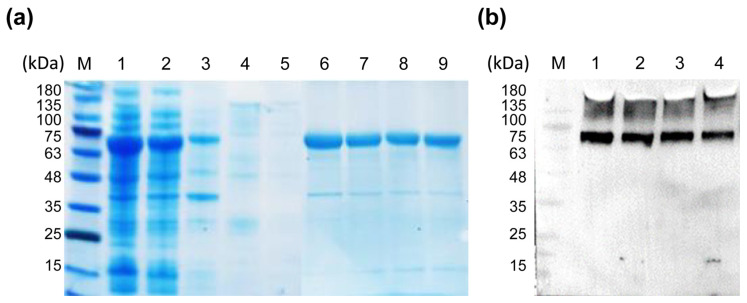
Purification of XXA tagged TMRPPSS2. (**a**) SDS–PAGE analysis of different samples taken during the purification process of the recombinant protein. Lane M: protein standard molecular weight. Lane 1: crude soluble cell lysate containing XXA–TMPRSS2. Lane 2: flow–through fraction from the Ni–NTA affinity Sepharose column. Lanes 3 to 5: protein sample eluted by low concentration of Imidazole (20 μM). Lanes 6 to 9: purified protein eluted by high concentration of Imidazole (100–300 μM) from one Ni–NTA affinity Sepharose column. (**b**) Lanes 1 to 4 purified XXA–TMPRSS2 proteins (corresponding to lanes 6 to 9 in panel (**a**) were analyzed by immunoblotting with anti–His antibody.

**Figure 4 ijms-24-10475-f004:**
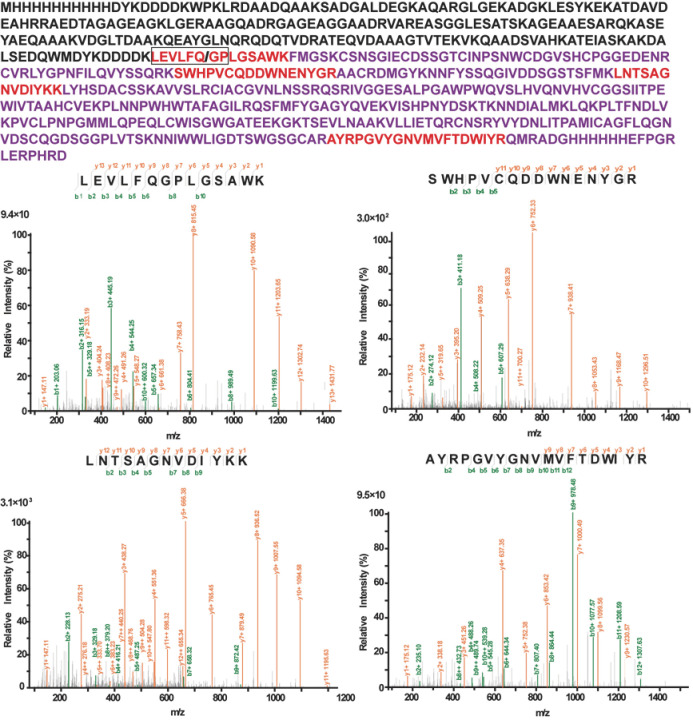
The MS–MS fragmentation spectra of four peptides selected from the peptide mass fingerprint (PMF) spectrum. The result analysis was performed from fragments of XXA–TMPRSS2 derived via trypsin digestion. Representative sequences coverage of these fragments was highlighted in red. The amino acid sequence corresponding to protein XXA was indicated in black, while the sequence of TMPRSS2 was shown in purple. An HRV 3C protease–specific recognition sequence LEVLFQ/GP was introduced into the recombinant protein between XXA and TMPRSS2.

**Figure 5 ijms-24-10475-f005:**
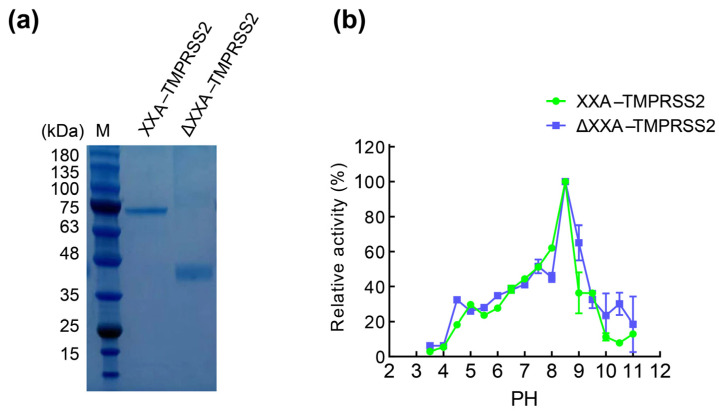
Effect of pH on the activity of XXA–TMPRSS2 and ΔXXA–TMPRSS2. (**a**): ΔXXA–TMPRSS2 was generated by the proteolytic cleavage of XXA–TMPRSS2 with HRV 3C Protease. (**b**): The influence of pH on the proteolytic activity of TMPRSS2 was shown. The reaction was carried out at different pH values ranging from 3.0 to 11.0 at room temperature with Boc–Gln–Ala–Arg–AMC as substrate. Values represent the mean ± standard deviation of three measurements. The relative activity of the enzyme was calculated considering the maximum activity as 100%.

**Figure 6 ijms-24-10475-f006:**
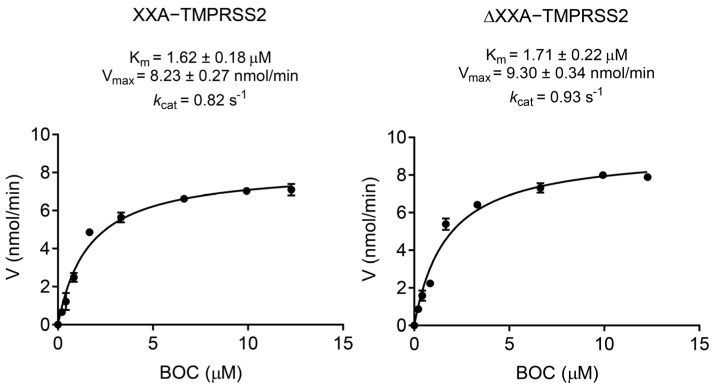
Kinetic parameter for XXA–TMPRSS2 and ΔXXA–TMPRSS2. The V_max_, *k*_cat_, and K_m_ were measured via plotting reaction velocity versus substrate Boc–Gln–Ala–Arg–AMC concentration at room temperature in pH 8.0. Values represent the mean ± standard deviation of three measurements.

**Figure 7 ijms-24-10475-f007:**
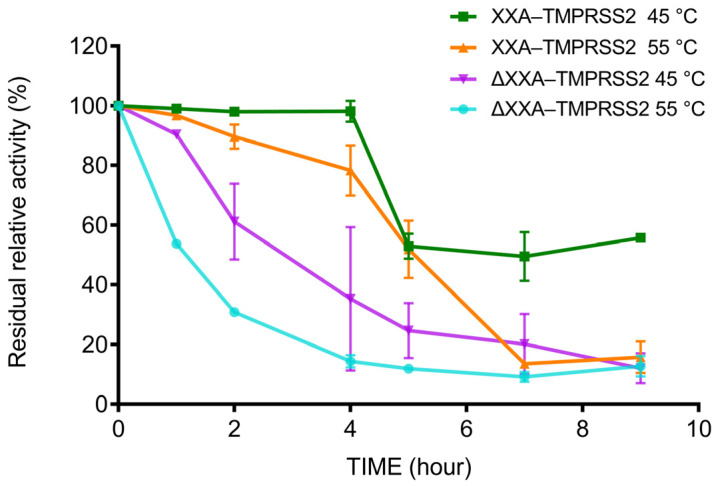
Thermostability assay of XXA–TMPRSS2 and ΔXXA–TMPRSS2. Thermostability of recombinant XXA–TMPRSS2 and ΔXXA–TMPRSS2 was determined by preincubating 1 mg/mL at 45 and 55 °C for designated time periods and then assaying the activity in reaction buffer at room temperature as described in “Materials and Methods”. The relative activity of the enzyme was calculated considering the maximum activity as 100%.

**Figure 8 ijms-24-10475-f008:**
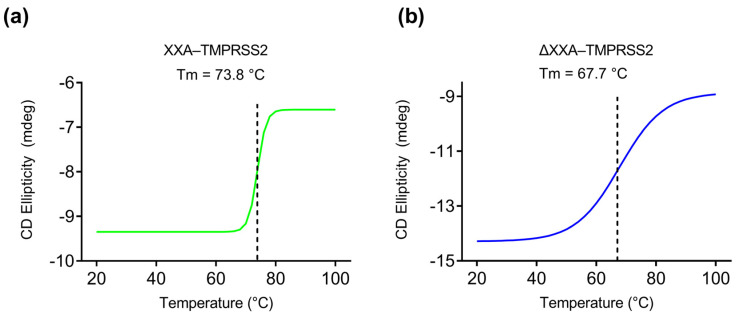
Protein denaturation analysis of XXA−TMPRSS2 and ΔXXA−TMPRSS2. (**a**) CD measurements at 220 nm of XXA−TMPRSS2 at different temperatures (20−100 °C); (**b**) CD measurements at 220 nm of ΔXXA−TMPRSS2 at different temperatures (20−100 °C). Dashed lines indicate the melting temperatures (Tm or the transition midpoint) of each enzyme. The Tm represents the temperature at which 50% of the protein is unfolded, which corresponds to the midpoint of the sigmoidal unfolding curve.

**Table 1 ijms-24-10475-t001:** Primers for recombinant plasmids cloning.

Plasmid	Inserted Gene	Primer F	Primer R
pET−His−TMPRSS2	His−TMPRSS2	TGGGTCGCGGATCCGCCTGGAAATTTATGGGC	GTGCGGCCGCAAGCTTGCCATCCGCGCGCATC
pET−MBP−TMPRSS2	MBP−TMPRSS2	AACAACCTCGGGGAATTCTGGAAATTTATGGGC	GGTGGTGGTGCTCGAGTTAGCCATCCGCGCGCATCTGAC
pET−SUMO−TMPRSS2	SUMO−TMPRSS2	CCAGGGGCCCGGATCCTGGAAATTTATGGGC	CGGCCGCAAGCTTGTTAGCCATCCGCGCGCA
pGEX−GST−TMPRSS2	GST−TMPRSS2	GGGGCCCCTGGGATCCTGGAAATTTATGGGC	AATTCTTAATGATGATGATGATGATGGCCATCCGCGCGC
pET−XXA–TMPRSS2	XXA–TMPRSS2	CAGGGGCCCGAATTCTGGAAATTTATGGGC	GGTGGTGGTGCTCGAGTTAGCCATCCGCGCGC

## Data Availability

The raw data used for all statistical analyses can be in the manuscript.

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
