# Peer review of "Improving Soluble Expression of SARS-CoV-2 Spike Priming Protease TMPRSS2 with an Artificial Fusing Protein"

_ijms, 2023, doi:10.3390/ijms241310475_

Round 1

Reviewer 1 Report

The authors of the manuscript use classical protein overexpression methods in a bacterial system (E.coli) to address the problem of abnormal folding/insolubility of expressed proteins. A novelty is the authors' proposed use of the retro-protein (misidentified by the authors as a peptide) XXA, as a sequence to facilitate correct folding and obtain a soluble protein (TMPRSS2). With this approach, the authors succeeded in obtaining a hybrid XXA-TMPRSS2-protein that is correctly folded/soluble (not going into inclusion bodies) and, most importantly, biologically (enzymatically) active. Overall, I have no major comments on the biotechnological part leading from plasmid construction to obtaining a functional protein.

The result obtained is valuable, containing the novelty of using the XXA protein to solve the problem of misfolding and protein insolubility during overexpression. I would recommend accepting the manuscript for publication after correcting errors and ambiguities.

 Before accepting the manuscript for publication, the authors should clarify/correct several issues described in a vague or incorrect manner.

My specific comments on the manuscript:

1. throughout the manuscript, the authors use the term peptide to describe the XXA sequence. In the source literature [22], the retro AXX protein sequence used by the authors (XXA), should count 192 aa. This is definitely not a peptide. It is a protein. And this is how it should be referred to starting from the title of the manuscript.

2 The use of the XXA sequence is new to this manuscript, yet the authors mention virtually nothing about it (except that they used it) in the introduction section. It would be appropriate to describe it briefly.

Results section:

Fig. 1. The photograph of the gel has defects in the lower part of the gel, however, this is not a substantively relevant part of the manuscript.

According to my cursory calculations, the XXA-TMPRSS2 hybrid should have a mass of about 62.3 kDa (42.8 kDa TMPRSS2+19.5 kDa XXA (according to [22]). Throughout the manuscript, the authors characterise this hybrid by assigning it a mass of 67 kDa (Fig. 2b,3a, 3b). Why such a difference?

Fig. 4. The MS spectra are too small and thus unreadable (markings, figures). The coverage of the full sequence of the hybrid protein by the identified 4 short peptides is not proof of the correctness of the sequence of the expressed protein. It is only an indication. 

In the XXA-TMPRSS2 hybrid sequence shown at the top of the figure, one would be tempted to mark the parts belonging to XXA and TMPRSS2. I suggest also marking the cut site/sequence for the HRV3c protease.

Why does the N-terminal part of the sequence presented at the top of Figure 4 (should correspond to XXA) not correspond to the one in the literature [22]?

 Kinetic parameters of XXA-TMPRSS2 p.6

the unit Vmax should be written as nmol/min, not nm or nM (this defect is both in the text and in the description of Figure 6)

Also, the value of Km should be expressed as the "Greek letter micro "M, not uM.

CD data p.7

Characterising the thermostability of a protein using the 220 nm point in a CD experiment makes sense if the dominant structure of the protein is helical. The authors nowhere mention whether this is the case. I suggest including a CD spectrum of the XXA-TMPRSS2 protein taken over the entire UV range in the supplements section.

Section 4 Materials and Methods

Section 4.5 Mass spectrometry

The composition given is for the eluent/solvent/mobile phase, not the HPLC 'buffer'. 98% of ACN is not standard procedure. Are you sure it is not 100% ACN? Also missing are column parameters (length, width, grain size) and eluent flow rate and phase A composition.

Supplements section

Plasmid diagrams are too small and thus unreadable.

the end of revision

Author Response

Reviewer: 1

Point 1: Throughout the manuscript, the authors use the term peptide to describe the XXA sequence. In the source literature [22], the retro AXX protein sequence used by the authors (XXA), should count 192 aa. This is definitely not a peptide. It is a protein. And this is how it should be referred to starting from the title of the manuscript.

Response 1: Thank you for bring up this issue. We totally agree that XXA should be referred to as a protein rather than a peptide. We apologize for our inaccurate statement and have made the necessary changes to the title and throughout the revised manuscript.

Point 2:The use of the XXA sequence is new to this manuscript, yet the authors mention virtually nothing about it (except that they used it) in the introduction section. It would be appropriate to describe it briefly.

Response 2: We appreciate the reviewer's suggestion to provide a full introduction to XXA before using it in the manuscript. In response, we added a new paragraph (Line 54-61) “XXA is a recently developed protein solubilizing fusion tag [15]. XXA has a re-versed amino acid sequence of the antifreeze protein from Chlorella sorokiniana (Sup-plementary Table 1), AXX, making it a retro-protein of AXX and also classifying as an artificial protein since its amino acid sequence does not exist naturally. With high sol-ubility and hydrophilicity, XXA exerts good properties in facilitating the refolding of inclusion bodies, and it outperforms other tags such as SUMO, MBP, and GST when tested with proteins like protease bdNEDP1 and nanobody NbALFA[15]”to describe XXA in detail, including its origin as a retro-protein of AXX and its amino acid sequence length of 192 residues that classifies it as a protein rather than a peptide.

Point 3: Fig. 1. The photograph of the gel has defects in the lower part of the gel, however, this is not a substantively relevant part of the manuscript.

Response 3: We agree with the reviewer that there are some abnormalities in the lower part of the gel. However, the portion corresponding to the expected molecular weight of TMPRSS2 (42 kDa) is intact and in good shape. This demonstrates the successful expression and purification of the protein, which is the focus of our study

.

Point 4:According to my cursory calculations, the XXA-TMPRSS2 hybrid should have a mass of about 62.3 kDa (42.8 kDa TMPRSS2+19.5 kDa XXA (according to [22]). Throughout the manuscript, the authors characterise this hybrid by assigning it a mass of 67 kDa (Fig. 2b,3a, 3b). Why such a difference?

Response 4: Thanks for bring up this issue. The addition of a HRV 3C site comprising 10 amino acids and a Flag label consisting of 10 amino acids between XXA protein and TMPRSS2, along with an 8*His tag added to the N-terminus of the XXA tag, resulted in an actual size of 67 kDa, which is larger than the theoretical size of 62.3 kD

Point 5:The MS spectra are too small and thus unreadable (markings, figures). The coverage of the full sequence of the hybrid protein by the identified 4 short peptides is not proof of the correctness of the sequence of the expressed protein. It is only an indication.

Response 5: Thank you for your comment. We apologize for any inconvenience caused by the small size of the MS spectra in our manuscript. In response, we have created a new graph with significantly improved quality, including larger size and enhanced readability.

Point 6:In the XXA-TMPRSS2 hybrid sequence shown at the top of the figure, one would be tempted to mark the parts belonging to XXA and TMPRSS2. I suggest also marking the cut site/sequence for the protease.

Response 6: Thanks for the suggestion.  We have made a figure with corresponding amendments. In the new figure the amino acid sequence of XXA was indicated in black, while the sequence of TMPRSS2 was shown in purple.  An HRV-3C protease specific recognition sequence LEVLFQ/GP was also highlighted and a slash (/) was used to indicated the cleavage site.

Point 7: Why does the N-terminal part of the sequence presented at the top of Figure 4 (should correspond to XXA) not correspond to the one in the literature [22]

Response 7: We apologize for the mistake made in Figure 4 where we inadvertently included the wrong sequence. We have made a new figure with necessary amendments.

Point 8:the unit Vmax should be written as nmol/min, not nm or nM (this defect is both in the text and in the description of Figure 6),Also, the value of Km should be expressed as the "Greek letter micro "M, not uM.

Response 8: We apologize for our mistake. We have corrected the corresponding mistakes in the revised manuscript. “Line 176、177,p6: the unit Vmax “ nM “has been changed to “nmol/min “;Line 176、177,p6: the value of Km the unit Vmax “ uM “has been changed to μM

Point 9: Characterising the thermostability of a protein using the 220 nm point in a CD experiment makes sense if the dominant structure of the protein is helical. The authors nowhere mention whether this is the case. I suggest including a CD spectrum of the XXA-TMPRSS2 protein taken over the entire UV range in the supplements section.

Response 9:Thanks for your valuable comments. The tertiary structure of TMPRSS2 (PDB: 7MEQ) indicates that the protein's predominant structure is α-helical. Therefore, we utilized the 220 nm point to assess the protein's thermostability.  The CD spectrum of the XXA-TMPRSS2 and ΔXXA-TMPRSS2 were captured over the entire UV range in Supplementary Figure 4.

Point 10: The composition given is for the eluent/solvent/mobile phase, not the HPLC 'buffer'. 98% of ACN is not standard procedure. Are you sure it is not 100% ACN? Also missing are column parameters (length, width, grain size) and eluent flow rate and phase A composition.

Response 10: Sorry, this is really my negligence. After carefully reviewing the laboratory record book, I discovered that some of the descriptions in my manuscript were not entirely accurate. I have revised the original text from” The digested peptides were separated on a nanoElute liquid chromatography system equipped with a C18-AQ reverse phase column in gradient of 5–95% HPLC buffer (0.2% formic acid in 98% acetonitrile, v/v).” to ”The digested peptides were separated within 30 min at a flow rate of 350 nL/min on a homemade column (25 cm×75 μm, 1.9 μm C18-AQ particles, Dr. Maisch). Solvent A was 0.1% (v/v) formic acid(FA), in H2O, and solvent B was  0.1% (v/v) FA, in 100% ACN. Peptides were separated with a gradient of 5–95% HPLC buffer B.” in Line326,p10.

Point 11: Plasmid diagrams are too small and thus unreadable.

Response 11: Thank you for your feedback. In response, we have created a new version of Supplementary Figure 2 with improved resolution to enhance its clarity and readability.

Reviewer 2 Report

This manuscript reports on the use of a recently invented tag protein named XXA (191 amino acids) for the expression in Escherichia coli BL21 of recombinant Transmembrane serine protease 2 (TMPRSS2). Compared to other protein tags used for expression in E. coli (SUMO, MBP and GST), the XXA tag provides recombinant TMPRSS2 mainly in soluble form. This is the first reported use of the XXA tag after its original description by Xie et al. (2022) (reference 22 in the manuscript). The interest of the manuscript relies on the known role of TMPRSS2 protease in priming the spike protein of SARS-CoV-2 to initiate cell infection, and on its being a good target for antiviral therapy. The expression of recombinant TMPRSS2 at high levels in soluble form will be advantageous for researchers in this field. The study is interesting and well written. However, some points can be improved.

MAJOR ISSUES

The sequence of XXA (the reverse sequence of the antifreeze protein AXX) is not shown in the manuscript. The reader would be thankful if this sequence is shown somewhere, or at least an explicit mention to where can it be found.

Line 35. Explain briefly what is the S2´ site.

Line 38. What is the meaning of “spike viral evasion”. This is not clear in reference 5.

Figure 2c. What are the black dots on top of the bars? Explain what is 100%. Error bars are missing.

Figure 4. State explicitly what is the sequence in the upper part of the figure. Only four peptides were identified? The four panels with partial MS spectra are practically useless. I suggest that, instead of this, a full comparison of the experimental and theoretical PMF should be shown.

Line 176-177. Vmax values do not coincide with those calculated in Figure 6. Vmax units are wrong: nm means nanometer, it should be nM (nanomolar).

Line 176-177. Since the proteins are of high purity, the authors should calculate kcat and kcat/Km.

Figure 6. Vmax units are wrong (nm should be nM). Km units are wrong (uM should be µM). Units of horizontal axes are wrong (they should be µM).

Figure 8. The different sigmoidal curves of both panels should be commented and their meaning explained.

Line 338. “2 µl of BOC-Gln-Ala-Arg-AMC”: concentrations should be stated.

Supplementary Figure 2. The plasmid drawings are illegible. Resolution should be improved

MINOR ISSUES

Line 59. Change “was” to “is”.

Figure 1. The red box is unnecessary since the theoretical molecular weight of His-TMPRSS2 is indicated by the black arrow.

Figure 2. Define TM.

Figure 5 and Figure 6. Define error bars.

Figure 7. The vertical axis label should be “Residual relative activity”

Line 202. Change “220 nM” to “220 nm”.

Line 339 and 342. The incubation period is duplicated (1 hour = 60 min)

Reference list. 14 and 18, numbers are duplicated

Reference list. 21 and 25, fist author name is incomplete

English language is fine. Minor editing required

Author Response

Reviewer2

Point 1: The sequence of XXA (the reverse sequence of the antifreeze protein AXX) is not shown in the manuscript. The reader would be thankful if this sequence is shown somewhere, or at least an explicit mention to where can it be found.

Response 1: Thanks for your advice. We have added the amino acid sequences of XXA and AXX in Supplementary Table 1

Point 2:Line 35.

Response 2:S2´ site is a cleavage site on the S2 subunit ,which is located upstream of the hydrophobic fusion peptide . TMPRSS2 cleaves the S2’ to trigger the exposure of the membrane fusion domain hydrophobic peptide (FP)  

Point 3:Line 38. What is the meaning of “spike viral evasion”. This is not clear in reference 5.

Response 3: Thank you for bringing this to our attention. We apologize for the inaccuracy in referring to "spike viral evasion" instead of "spike protein mediated viral cell evasion". To ensure accuracy in our manuscript, we have made the necessary corrections in the revised version.

 Point 4:Figure 2c. What are the black dots on top of the bars? Explain what is 100%. Error bars are missing.

Response 4: The black dots on top of the bars in the chart represent the data for each experimental condition, while the bar chart represents the average of the two experiments. The relative activity of the enzyme was calculated by considering the maximum activity as 100%.

Point 5:Figure 4. State explicitly what is the sequence in the upper part of the figure. Only four peptides were identified? The four panels with partial MS spectra are practically useless. I suggest that, instead of this, a full comparison of the experimental and theoretical PMF should be shown.

Response 4: Thank you for your suggestion. We have created a new version of Figure 4 that includes several improvements to enhance clarity and readability. We have indicated XXA and TMPRSS2 with different colors and highlighted the HRV-3C protease cleavage site (LEVLFQ/GP). The MS spectra of the four fragments provide strong evidence for the correct expression of TMPRSS2, as each of the sequences is uniquely corresponding to TMPRSS2. The presence of 4 of the sequences in one MS analysis makes it highly unlikely that the sample was contaminated with other proteins. Therefore, we can confidently conclude that TMPRSS2 was expressed as intended.

Point 6: Line 176-177. Vmax values do not coincide with those calculated in Figure 6. Vmax units are wrong: nm means nanometer, it should be nM (nanomolar).

Response 6: Sorry, this is really my negligence,Line 176-177, Vmax values is consistent with those calculated in Figure 6 after modification. Line 176-177,p6: the unit Vmax “ nM “has been changed to “nmol/min “

Point 7:Line 176-177. Since the proteins are of high purity, the authors should calculate kcat and kcat/Km.

Response 7: Line 176-177,p6:The values of kcat and kcat/Km are added in the paper.

Point 8:Figure 6. Vmax units are wrong (nm should be nM). Km units are wrong (uM should be µM). Units of horizontal axes are wrong (they should be µM).

Response 8: Line 176-177,p6:the unit Vmax “ nM “has been changed to “nmol/min “,the Km units “uM” has been changed to” µM”

Point 9:Figure 8. The different sigmoidal curves of both panels should be commented and their meaning explained

Response 8:Upon elevating the temperature, both proteins underwent denaturation and lost their secondary structure, as evidenced by changes in the ellipticity measured at wavelength of 220 nm in our study. We have also added a new CD spectrum of XXA-TMPRSS2 and ΔXXA-TMPRSS2 protein in the entire UV range in Supplementary Figure 4 to should the whole protein structure of the two proteins.

Point 10: Line 338. “2 µl of BOC-Gln-Ala-Arg-AMC”: concentrations should be stated.

Response 10: Line 338,p11:”2 µl of BOC-Gln-Ala-Arg-AMC”: concentrations have been added -10μΜ.

Point 11:Supplementary Figure 2. The plasmid drawings are illegible. Resolution should be improved

Response 11: We have updated the resolution of the plasmid drawings included in our supplementary materials.

Point 12: Line 59. Change “was” to “is”

Response 12: Line 59,p2:” was” has been changed to “ is”.

Point 13:  Figure 1. The red box is unnecessary since the theoretical molecular weight of His-TMPRSS2 is indicated by the black arrow.

Response 13:  The black arrow has been removed in new Fgure 1

Point 14: Figure 2. Define TM.

Response 14: Tm(melting temperature) : The temperature corresponding to the midpoint of the protein transition from folding to unfolding is called Tm. We have added the definition of Tm into Line213,p7.

Point 15: Figure 5 and Figure 6. Define error bars(172)

Response 15: The definition of error bars is that values represent the mean ± standard deviation of two measurements.I have added the definition of error bars into Line 172,p6 :Figure 5 and Line 184,p6 :Figure 6

Point 16:Figure 7. The vertical axis label should be “Residual relative activity”

Response 16: Figure 7. The vertical axis label has been changed to “Residual relative activity”

Point 17:Line 202. Change “220 nM” to “220 nm”

Response 17: Line202,p7: “220 nM” has been changed to “220 nm”

Point 18:Line 339 and 342. The incubation period is duplicated (1 hour = 60 min)

Response 18:The first incubation of 1 hour means that the reaction system is kept at room temperature, and fluorescence is not measured during this process. The second incubation of 60 min means that the fluorescence was measured within 60 min after the first incubation.

Point 19: Reference list. 14 and 18, numbers are duplicated

Response 19: The duplicated number of reference list. 14 and 18 have been deleted.

Point 20: Reference list. 21 and 25, fist author name is incomplete

Response 20: The first author’s name has been completed on page 11 The first author’s name of the reference list. 21 and 25 have been completed

Round 2

Reviewer 2 Report

Part of the queries raised in my first report have been attended, but not all, and in some cases the solution given makes manifest new problems.

Below I use as reference the numbering of points in the cover letter of the revised manuscript.

Point 1. Supplementary Table 1 is mentioned in the cover letter and in the revised manuscript, but it has NOT been submitted.

Point 5. Figure 4. The Figure has been changed, as well as the selection of peptides shown in the four panels. I agree that the identification of the four sequence is sufficient for the intended use of the data. However the description of Figure 4 as a “peptide mass fingerprint” (PMF) is wrong. In fact the figure shows the MS-MS fragmentation spectra of four peptides selected from the PMF. This should be taken into account also in Methods Section 4.6.

      In addition, the legend to Figure 4 is shown before the Figure, not after it.

Point 7. Lines 189-192 of the revised manuscript and the revised Figure 6. The symbol for second is “s” (lowercase), and the symbol for catalytic constant is kcat (lowercase k) NOT Kcat.

Point 9. Figure 8 and lines 215-226. The requested comment and explanation for the different sigmoidal curves has not been attended, and I still believe it should be.

      A new Supplementary Figure 4 is mentioned in the cover letter and in the revised manuscript ((line 218), but it has NOT been submitted.

Point 11. The plasmid drawings of Supplementary Figure 2 are the same as in the original version. The resolution has NOT been updated, and the drawings continue being illegible.

Point 14. Figure 2. The definition of “TM” in Figure 2 was requested, but the authors have defined “Tm” in Figure 8. The definition of TM in Figure 2 is still needed.

English is fine. Minor editing required

Author Response

Dear Editors and Reviewers,

We would like to express our sincere appreciation for the numerous valuable comments and suggestions you provided regarding this manuscript. Your constructive feedback has been thoroughly considered, resulting in a comprehensive revision of the manuscript. We have incorporated the necessary modifications accordingly, and the detailed corrections are provided below. We kindly request you to review the revised manuscript and refer to the responses below for further details.

Thank you in advance for your time and attention.

Point 1. Supplementary Table 1 is mentioned in the cover letter and in the revised manuscript, but it has NOT been submitted the Supplementary Table 1

Response 1: We apologize for the negligence. We have now uploaded all the required supplementary files as requested.

Point 2. Figure 4. The Figure has been changed, as well as the selection of peptides shown in the four panels. I agree that the identification of the four sequence is sufficient for the intended use of the data. However the description of Figure 4 as a “peptide mass fingerprint” (PMF) is wrong. In fact the figure shows the MS-MS fragmentation spectra of four peptides selected from the PMF. This should be taken into account also in Methods Section 4.6   In addition, the legend to Figure 4 is shown before the Figure, not after it.

Response 2:Thank you for pointing out the mistakes. After consulting with our MS experts, we have made the necessary modifications to the description of the MS analysis as well as the procedures in the revised manuscript.

Additionally, I have rearranged Figure 4 and its legend.

Point 3: Lines 189-192 of the revised manuscript and the revised Figure 6. The symbol for second is “s” (lowercase), and the symbol for catalytic constant is kcat (lowercase k) NOT Kcat.

Response 3: Thanks for point out the typos. We edited the manuscript and made the necessary modifications: Lines 189-192,p7 and Figure 6.the “S” has been changed to “s”, and “kcat” has been changed to “kcat”.

Point 9. Figure 8 and lines 215-226. The requested comment and explanation for the different sigmoidal curves has not been attended, and I still believe it should be.

Response 9: We apologize for not addressing the questions in our previous revision. We have now rectified this by providing an updated version of the section that offers a more detailed explanation of the protein denaturation curve

Point 11. The plasmid drawings of Supplementary Figure 2 are the same as in the original version. The resolution has NOT been updated, and the drawings continue being illegible.

Response 11: We apologize for our negligence. We have thoroughly checked the manuscript and uploaded revised Figures with improved quality. 

Point 14. Figure 2. The definition of “TM” in Figure 2 was requested, but the authors have defined “Tm” in Figure 8. The definition of TM in Figure 2 is still needed

Response 14: Thank you for bringing this to our attention. In Figure 2, "TM" refers to TMPRSS2, and we have now clarified this in the manuscript. Additionally, in Figure 8, "Tm" represents the melting temperature, which signifies the temperature at which the protein is unfolded by 50%. Tm serves as a metric for the thermostability of a protein. We have made the necessary amendments in the legend of Figure 2, the materials and methods section, as well as the manuscript itself to provide a clear definition of Tm.

Round 3

Reviewer 2 Report

The queries raised in my second review have been satisfactorily attended, except that no supplementary material has been loaded in this occasion. Therefore, I could not check Supplementary Table 1, nor the plasmid drawings of Supplementary Figure 2.

Minor editing of English needed